# Apparent False Lateralization of Seizure Onset by Scalp EEG in Temporal Lobe Epilepsy Associated with Cerebral Cavernous Malformation: A Case Report and Overview

**DOI:** 10.3390/brainsci10090584

**Published:** 2020-08-24

**Authors:** Mariana Gaviria Carrillo, Jonathan López, Jesús H. Rodríguez Q., Ivan Gaona, Gloria Ortiz-Guerrero, Mauricio O. Nava-Mesa

**Affiliations:** 1Department of Neurology, Fundación Cardioinfantil (FCI), Bogotá 110131, Colombia; marigaviria8@hotmail.com (M.G.C.); jrodriguez@cardioinfantil.org (J.H.R.Q.); ivangaona@hotmail.com (I.G.); 2Escuela de Medicina y Ciencias de la Salud, GI en Neurociencias-NeURos, Universidad del Rosario, Bogotá 111221, Colombia; 3Department of Neurology, Universidad del Sinú, Cartagena 130001, Colombia; jonathanlopez244@hotmail.com; 4Department of Neurology, Hospital Universitario Mayor—Mederi, Bogotá 111411, Colombia; 5Department of Neurology, University of Kansas Medical Center, Kansas City, KS 66160, USA; gloriaortiz9024@gmail.com

**Keywords:** cavernoma, false lateralization, epilepsy surgery, scalp EEG, refractory epilepsy

## Abstract

False lateralization of ictal onset by scalp electroencephalogram (EEG) is an infrequent entity that has been reported in patients with mesial temporal lobe epilepsy associated with hippocampal sclerosis (HS). In these cases, a tendency for rapid seizures that spread through the frontal-limbic system and hippocampal commissural pathways to the contralateral hemisphere has been proposed. Cerebral cavernous malformations (CCMs), which constitute a collection of abnormally configured small blood vessels with irregular structures, is a well-defined epilepsy-associated pathology. Their primary association with seizures might be explained either as a result of physiological changes affecting the cerebral cortex immediately surrounding the CCM (an epileptogenic mechanism that is relevant for both, temporal and extratemporal lesions) or as a result of promoting epileptogenicity in remote but anatomo-functionally connected brain regions, a mechanism that is particularly relevant for temporal lobe lesions. To date, there have been only two publications on falsely lateralizing ictal onsets by EEG in temporal cavernoma, but not in other regions. Here, we report a rare case of apparent false lateralization of ictal onset by scalp EEG in a patient with a left medial frontal gyrus cavernoma (supplementary motor area), and discuss some relevant pathophysiological mechanisms of false lateralization.

## 1. Introduction

Infrequently, seizures may be falsely lateralized to the contralateral hemisphere based on ictal scalp electroencephalogram (EEG) recordings. False lateralization of ictal onset by scalp EEG has been reported in patients with severe hippocampal sclerosis (HS) and cortical atrophy [1,2] as well as lesions in other supratentorial regions like the superior temporal gyrus, hippocampus, and neocortical temporal lobe [3] (Table 1). Mintzer et al. found that 4.7% of their cohort with severe HS had ictal onset in the normal temporal lobe on scalp ictal EEGs, but depth electrode recordings then showed ictal onset in the mesial temporal lobe ipsilateral to the imaging abnormality [1,4].

Cerebral cavernous malformations (CCMs), also known as cavernomas, cavernous angiomas, or cavernous hemangiomas, occur in up to 0.5% of the general population (asymptomatic prevalence of 1 in 625) and constitute between 5% and 10% of all brain and spine vascular abnormalities [5,6]. According to several reports, patients with CCMs, along with other vascular abnormalities (e.g., arteriovenous malformations—AVM) have a higher risk of suffering a seizure according to hospital-based studies (between 30% and 47%) [7]. In contrast, population-based studies reported that around 25% of patients with these vascular anomalies presented with epilepsy [7,8]. Correspondingly, in patients with CCMs without bleeding or neurological impairment, the five-year risk of a first seizure is around 4% [9]. In the case of AVMs, this risk increases after intracranial hemorrhage or focal neurologic deficit (from 8% to 23%) [6]. The occurrence of epilepsy due to CCMs is associated with lesion multiplicity and cortical location. In fact, superficial supratentorial cavernomas are associated with seizures in up to 50–60% of patients, most of whom subsequently develop epilepsy. Other factors associated with seizures in patients with CCMs include the brain region (cortical more than subcortical involvement) and type of cortex affected (mesiotemporal more than arquicortical involvement) [9].

Only two cases of false lateralization by scalp EEG have been described in patients with temporal lobe cavernomas, specifically in the right superior temporal gyrus and the hippocampus [10,11] (Table 1). Here, we report a rare case with apparent false lateralization of ictal onset by scalp EEG in a patient with a left medial frontal gyrus cavernoma. We also discuss the clinical and pathophysiological mechanisms of cavernous malformation in the development of recurrent seizures and false lateralization.
brainsci-10-00584-t001_Table 1Table 1Summary of demographic, neuroimaging, and EEG findings of false lateralization cases in the literature.Age (Years)-SexClinical Features and Ictal SemiologyMRI (Structural Findings)Functional NeuroimagingScalp EEGIntracraneal Ictal EEG False Lateralization HypothesisRef.56, female Focal onset, awareness impairment, non-motor features. Normal No area of hypo metabolismInterictal: sharp waves from both temporal head regions.Ictal: rhythmic 5-6 ictal activity from the left temporal head region. Ictal seizure onset in the right temporal lobe. 1.5 s after seizure onset on the right: gamma activity in the left mesial temporal and the left lateral temporal electrodes.Temporal neocortex epilepsy: rapidly involve the contralateral mesolimbic structures; spread through the hippocampal commissures. Initial low voltage fast activity in the right temporal was undetectable on scalp EEG.[1]25, femaleFocal onset, awareness impairment, non-motor features and motor features (turn her head to the left, chewing, and hyperkinetic movements) Atrophic process in the left hemisphere Not reportedInterictal epileptiform in both inferomesial temporal regions, left side predominance. Ictal: right temporal region onset, two seconds later it spread to the contralateral hemisphere, amplitude predominance was right Seizure onset started in the left temporal. 36 s after the seizure onset, the ictal discharge spread to the right temporal lobeCerebral damage reduces the voltage of the ictal discharge on the side of the lesion and higher amplitude in the normal hemisphere.[2]21, male Unknown onset seizure with motor activity. Focal onset, and impairment awareness, motor features (automatism), and non-motor features (epigastralgic sensation)Left hemisphere atrophy Not reported Interictal. sphenoidal EEG: bilateral temporal, independent sharp waves with left sided predominance Ictal: discharge arise from the right frontotemporal regions, predominance over the right temporal region, 12 s after the seizure onset, ictal discharge spread to the contralateral hemisphere. Onset of the ictal discharge in the left temporal lobe Cerebral damage reduces the voltage of the ictal discharge on the side of the lesion and higher amplitude in the normal hemisphere.[2]30, male Focal onset, and impairment awareness , non motor features (fear) and motor features (automatism)Enlarged left lateral ventricle,three small areas of hypo density in the left posterior limb of theinternal capsule
Bitemporal independent, interictal sharp waves with right-sided predominance.Ictal EEG: seizure onset in the right temporal lobe. Seizure discharge started in the left temporal lobe. 2–3 s after seizure onset ictal discharge spread to the right temporal lobe also spread to both frontal regions. Cerebral damage reduces the voltage of the ictal discharge on the side of the lesion and higher amplitude in the normal hemisphere. [2]39, female1. Focal onset, and impairment awareness , non motor features (epigastric sensation) 2. Focal onset, and impairment awareness and bilateral progression Atrophy of the left cerebral hemisphere.Interictal Ictal ECD- SPECT: left hemispheric hypo perfusion and IMZ SPECT revealed slight hypo accumulation in the left hippocampusIctal: Right temporal side focusInterictal subdural EEG: independently in the left hippocampus region and also the right hippocampus region with spread to the lateral temporal regions.Reduced epileptiform activity in the atrophic hemisphere. Scalp EEG and MEG propagated electricals signals to the right side are easier to detect than signs to the left side. [3]23, male Focal onset, epigastric sensation, impaired awareness, and motor features (automatisms)Severe left hippocampal sclerosis Left temporal hypometabolism Interictal scalp EEG: mild left temporal slow, left midpost, temporal spikes. Ictal scalp: 4 right temporal, 1 left temporalLeft mesial temporal“Burned-out hippocampus” syndrome with atypical spread of ictal discharges. Not enough neocortical neurons to produce a visible scalp discharge is not accomplished until propagation of the seizure to the contralateral temporal lobe.[4]48, male Focal onset, and motor features (automatisms)Severe left hippocampal sclerosisNot reported Interictal scalp EEG: Bitemporal spikes Ictal scalp: 5 right temporal, 2 bitemporal Left hippocampus/amygdala and lateral temporal spikes. “Burned-out hippocampus” with atypical spread of ictal discharges. Rapid bilateral spread.[4]38, female Bilateral tonic-clonic seizures, impaired awareness and non-motor components (panic attacks). Recalled memory, epigastric sensation.Mild right hippocampal atrophy PET: Mild right temporal hypo metabolism Interictal: mild generalized slowing, right anterior temporal spikes, rare left anterior temporal sharps Ictal scalp: theta frequency rhythmic discharge in the left temporal region.Frequent spikes in the right mesial entorhinal cortex. Right anterior hippocampus and amygdala. Propagation through the dorsal hippocampal commissure was faster than that to the overlying ipsilateral neocortex, thereby producing falsely discordant ictal scalp EEG recordings. Location and extent of the epileptogenic region: a more posterior hippocampal propagation pattern.[4]37, femaleFocal onset, impaired awareness and bilateral tonic-clonic seizures. Lightheadedness, staring, fumbling.Severe right hippocampal atrophy PET: hypo metabolism: mild right temporal Interictal infrequent left temporal sharp wavesIctal scalp: left temporal onsetInitial seizures (slowing and spikes and sharp waves)bitemporal,later ones righttemporal“Burned-out hippocampus” with atypical spread of ictal discharges. Interhemispheric propagation.Propagation through the dorsal hippocampal commissure was faster than that to the overlying ipsilateral neocortex, thereby producing falsely discordant ictal scalp EEG recordings[4]27, maleFocal onset, impaired awareness and motor features (bimanual automatisms), epigastric sensation. Severe left hippocampal atrophy Not performed Interictal: left mid temporal spikes Ictal: 3 right midtemporal onset, 1 bitemporal onsetSlowing and frequent spikes in the left hippocampus, some spikes in the right hippocampus. “Burned-out hippocampus” with atypical spread of ictal discharges.[4]12, maleAuditory aura, bimanual and oralautomatisms, impaired awareness and non-motor features (déjà vu)Cavernous malformation in the right superior temporal gyrusF-FDG-PET focal hypo metabolism on the right, left temporal hypo metabolism. SPECT ictal and interictal: focal hypo perfusion at the location of the cavernoma. Interictal: left temporal hypo perfusion Ictal: hyper perfusion on the left temporal lobe.Abnormal perfusion of the left temporal lobe and along the caudal margin of the right temporal cavernoma.MEG: spikes sources around the cavernoma, but also contralateral in the left miid lateral temporal region. Left frontal and posterior head quadrant spikes and a single electro clinical seizure originating in the left frontal head region. Not performed in the presurgical stateDistinct site of epileptogenesis contralateral to the cavernoma (two epileptogenic foci) orrapid contralateral spread of epileptogenic activity.[10]34, femaleFocal onset, memory deficits, impaired awareness, non motor (“weird feelings”) and motor features (automatisms). Cavernous angioma in the left medial temporal lobe Not performedInterictal: epileptiform patterns (right more than left)Ictal: two seizures with a right temporal origin and two of less certain onset Right mesial temporal onset (subdural strip electrodes)Focal neurochemical abnormalities that coincide with ictal onsets.[11]33, not reported. Not seizure description. Left, fronto-temporal dysfunction in the NPT and Wada test.Left hippocampal sclerosis Not performed Interictal: epileptiform discharges in the left temporal region.Ictal: 2 ictal onsets in the left, 4 seizures onset in the right temporal region.Ictal onset in the left hippocampus.Seizure spreading to contralateral neocortex through functional commissural connections (i.e., dorsal hippocampal commissure)Rapid contralateral spread by activation of frontal limbic pathways.[12]27, not reported Not seizure description. Left, temporal dysfunction in the NPT and Wada test.Left hippocampal sclerosisNot performedInterictal: epileptiform discharges in the left side.Ictal: 2 ictal seizure onset in the right temporal regionIctal onset in the left hippocampus.Seizure spreading to contralateral neocortex through functional commissural connections (i.e., dorsal hippocampal commissure)Rapid contralateral spread by activation of frontal limbic pathways.[12]25, not reportedNot clinical informationLeft hippocampal sclerosisNot performedInterictal: epileptiform discharges in the left side.Ictal: 3 ictal seizure onset in the right sideIctal onset in the left anterior inferior–mesial temporal region.Seizure spreading to contralateral neocortex through functional commissural connections (i.e., dorsal hippocampal commissure)Rapid contralateral spread by activation of frontal limbic pathways.[12]33, not reportedNot seizure description. Right, temporal dysfunction in the NPT and Wada test.Right hippocampal sclerosisNot informationInterictal: epileptiform discharges in the right side.Ictal: 4 ictal seizure onset in the left temporal regionIctal onset: right hippocampus Seizure spreading to contralateral neocortex through functional commissural connections (i.e., dorsal hippocampal commissure)Rapid contralateral spread by activation of frontal limbic pathways.[12]45, maleFocal onset, impaired awareness, non-motor features (déjà vu and periodic nausea). Impairment of both verbal and visual memory in the NPT.Several bilateral scattered white matter hyper intensities,most prominent in the left frontal lobe.Decreased glucose uptake in the left temporallobe as compared with the right temporal lobe in both lateral and mesial cortical regions.Onset in the right amygdala, althoughone subclinical seizure began in the left temporallobe.EEG from the right amygdala showedcontinuous low-amplitude fast activity (>13 Hz) forextended times; at other times, this pattern was entirelyabsent. The fast activity was believed to beepileptic, although it was asymptomatic. Severaldiscrete subclinical seizures with a different electrographicpattern originated in the right amygdala,later spreading to the right hippocampus.Abnormal increase in temporal lobe metabolism.[13]27, femaleFocal onset, aware, non motor features (unpleasant smell and taste, salivation, notedstrange feelings in her face or body) Dilated right temporal ventricular horn with questionabledecreased hippocampal volumeLess FDGuptake in the left temporal lobe than in the righttemporal lobe, mainly in lateral temporal cortex.Right sphenoidalspikes and intermittent arrhythmic right sphenoidaltheta in wakefulness and sleep.All arose from the right amygdala. The EEG also showed frequent (severalper second) sharp waves and occasional subclinicalelectrographic seizures in the right amygdala.Sharp waves amplify neuronal activity in temporal lobe neocortex through amygdalocortical connections, possibly increasing lateral temporal cortical metabolism. [13]47, maleFocal onset, awareness impairment, non-motor features (déjà vu)Left hippocampal atrophy FDG PET: decreased uptake in the left temporal region Ictal EEG: Left temporal (1 min after the clinical onset of the seizure) SDE: seizure onset in the right lateral temporal region before spreading to the let lateral temporal SDE in 4 to 10 s.SEEG: Regional onset seizure arising from the left SEEG electrodes suggesting a left temporal onsetRapid contralateral spread through the hippocampal commissure or frontal limbic pathways  [14] 45, female 1. Unknown onset seizure with motor activity 2. Focal onset, awareness impairment, non motor features3. Focal onset, awareness impairment, and progression to bilateral tonic–clonic seizuresNormal FDG PET: normal. Interictal: Left temporal epileptiform discharges. Ictal recordings: no localizing and no lateralizing SEEG electrodes: seizures arising from the left temporal electrode, then spreading to the contralateral electrode first, followed 5 to 20 s later byspread to the contralateral temporal SDE; the ipsilateralSDE recordings showed no changeRapid contralateral spread through the hippocampal commissure or frontal limbic pathways [14]Abbreviations: ECD: ethyl cysteinate dimer; EEG: electroencephalogram; FDG: [‘8F] fluorodeoxyglucose; IMZ: I-123 iomazenil; MEG: Magnetoencephalography; MRI: magnetic resonance imaging; NPT: neuropsychological test; PET: positron emission tomography; SEEG: stereotactic depth EEG; SDE: subdural electrodes; SPECT: single-photon emission computed tomography.

## 2. Case Presentation

The patient was a 53-year-old right-handed female who presented an unknown onset seizure with motor activity (tonic- clonic) at the age of 17 years. Personal history for other neurological conditions was negative. Epilepsy risk factors, including febrile seizures, head trauma, central nervous system (CNS) tumor, and family history, were also negative. The patient denied smoking, drinking alcohol, or using illicit drugs. The neurological exam was normal, and she scored 26/30 on the Montreal Cognitive Assessment (MOCA) test. Interestingly, she did not have more seizures until she was 26 years old. This last seizure had a focal onset, with impaired awareness, and a motor component that involved manual automatism, blinking, cephalic version to the right, and asymmetric tonic limb posturing with progression to bilateral tonic-clonic seizures.

Her seizures worsened progressively due to increased frequency and severity, with an ictal frequency of five per month. She was on carbamazepine 400 mg daily, levetiracetam 3000 mg daily, lacosamide 250 mg daily, and lamotrogine 100 mg daily with poor response. Then, she was referred to our hospital (Hospital Universitario Mayor—Méderi) to characterize her seizures. A long-term (72 h) video EEG with scalp electrodes was performed with a 50% reduction of the antiepileptic drug (AED) doses. The video EEG recorded 14 seizures in total. She presented two different types of seizures with different semiology: the first type was a seizure with a focal onset, impaired awareness, and non-motor components such as behavior arrest, cognitive impairment due to disorientation and emotional seizures (psychomotor agitation, fear, and anxiety) and eye blinking. The second one was a focal onset seizure with impaired awareness, motor components such as eye blinking, cephalic version to the right, dystonic extension posture in the right arm (Figure 4 sign), and finally a progression to bilateral tonic-clonic seizure. After video EEG was completed, she was restarted on the full dose of AEDs, achieving seizure control within the next 24 h.

The scalp EEG showed interictal rhythmic spike activity in the left mesial temporal region maximal at the T3 electrode. A Left F7-T3 spike with intermittent fronto-central delta slowing was found. A right interictal temporal spike was also seen at the end of the recording (Figure 1). Ictal scalp EEG showed 4–5 Hz rhythmic activity that involved the right frontotemporal region. It spread to the contralateral hemisphere, where it was associated with epileptiform activity (Figure 2 and Figure 3).

A computed tomography (CT) scan showed a left frontal parasagittal lesion with calcifications without mass effect. Then, a 1.5 Tesla brain magnetic resonance image (MRI) was obtained and confirmed the presence of a 9 × 6 mm left frontal parasagittal lesion with the so-called ‘popcorn’ appearance. The nodule showed smooth, thin, and hypointense borders, with a homogeneously hyperintense core on the T2W (weighted) images. The T2 gradient echo (GRE) images defined the hemosiderin component of the lesion, also known as ‘the blooming effect’. It did not enhance on T1W post-gadolinium images (Figure 4). Positron emission tomography (PET) brain imaging using [18] FDG fluorodeoxyglucose showed a pseudonodular lesion with internal calcifications of 9 × 6 mm approximately without radiopharmaceutical uptake (ametabolic). No abnormal perfusion in the surrounding area next to the cavernoma was found. An acceptable distribution and radiofrequency uptake by other brain structures, including the temporal lobes, was shown (Figure 5).

In summary, the present case characterized by ictal semiology and onset in the left hemisphere matched to the side of the cavernoma localization. However, the scalp EEG showed a right onset and fast spread to the contralateral hemisphere, meaning there was conflicting evidence regarding lateralization.

### Outcome and Follow Up

According to several guidelines for cavernoma-related epilepsy management [9,15], surgical resection of the cavernoma was offered to the patient by our epilepsy group. However, the patient declined the procedure for personal reasons, and AED therapy was restarted. Medication dosage was adjusted (levetiracetam 2000 mg daily and lamotrigine 400 mg daily) due to the development of somnolence and drowsiness, resulting in partial control of the seizures.

## 3. Discussion

This case illustrates a patient with different types of seizures, some of them with lateralizing and localizing features. The second type seizure of our patient was characterized by head version to the right, followed by asymmetric tonic limb posturing and bilateral progression to tonic-clonic seizures, which localized the symptomatogenic zone to the left side. When the head version is the first and predominant symptom of a seizure, it usually helps to localize the epileptogenic area to the contralateral side as this is true in around 90% of cases [16,17]. If the seizure progresses to a bilateral tonic-clonic seizure, the forced head deviation would be caused by the activation of the frontal eye field and motor areas, anterior to the precentral gyrus and close to the left cerebral cavernous malformation (CCM) [17]. The right elbow assumes an extended position, and the left elbow flexes over the chest, assuming an asymmetric tonic limb posturing. This sign provides correct lateralization in most cases, in which the extended elbow is contralateral to the epileptogenic hemisphere [16]. According to the ictal semiology in our patient, the epileptogenic zone was identified in the left side, matching the CCM location. However, neither a scalp EEG nor functional tests were concordant.

Functional connections between the frontal lobe and hippocampus, and later rapid spread from commissural pathways between both hippocampi may underlie false lateralization. Indeed, rapid contralateral propagation of seizures by activation of temporal and frontal limbic pathways has been proposed in several studies [12,18,19]. Another hypothesis suggests that false lateralization by scalp EEG might be caused by severe tissue damage induced by inflammation and reactive gliosis, which leads to a reduction in the voltage on the side of the lesion. Likewise, in patients with temporal lobe epilepsy and focal brain lesion, the epileptiform discharges could be reduced and difficult to detect by extracranial EEG [1,2]. However, in the present case, no radiologic evidence of encephalomalacia surrounding the CCM was found. Finally, electrical interference caused by anatomical components such as meninges, skull, or scalp must also be considered in cases of false lateralization as it can affect the detection of the epileptic focus [20] (a summary of false lateralization hypothesis are included in Table 1).

Another possible explanation of false lateralization may be the presence of two epileptogenic loci. Correspondingly, there are some reports of coexistence of cavernoma and cortical dysplasia in patients with refractory epilepsy [21,22,23]. Some authors consider cavernomas as congenital abnormalities, similar to neurodevelopmental brain lesions (NDLs) such as cortical dysplasia and glioneuronal tumors [24]. Therefore, the exclusion of dual pathology, as well as the study of other vascular malformations are mandatory [25,26]. In a retrospective study, patients with cavernomas and NDL exhibit similar frequencies of sporadic spikes, as well as continuous spikes, burst, and recruiting spikes in the neocortical region [24]. In the same study, it is reported that the absence of coincident bursts was associated with high microglia density in patients with cavernomas, aiding to inhibit epileptogenic discharges in this area. In our case, continuous spikes were recorded in most of the EEG studies associated with occasional burst and recruiting spikes. This suggests a moderate microglia density. However, we did not have neuropathological studies for this patient.

Complementary to ictal semiology, invasive methods may help to reduce the probability of false lateralization. In fact, a retrospective study in patients with refractory focal epilepsy who were implanted with bilateral depth electrodes found that in 60% of cases, the scalp EEGs were discordant with intracranial EEG recordings: 27% of the scalp EEGs falsely lateralized the onset, and 33% suggested bilateral independent or single ictal onset [20]. However, advances in anatomic and functional neuroimaging (i.e., 18F-FDG PET and single photon emission computed tomography SPECT) have reduced the need for invasive monitoring in certain patients with focal epilepsy [27,28]. Although functional neuroimaging has shown several metabolic and perfusion changes in patients with refractory and focal epilepsy [13,27], in our clinical case the metabolism surrounding the lesion was unaffected according to the PET scan. Usually, it is expected to find increases in brain metabolism and cerebral blood flow in the epileptogenic focus. However, functional neuroimaging, such as SPECT, has a low sensitivity and high false-positive rate depending on the time the tracer injection is administered [29]. Although intracranial EEG is the gold standard to guide an epileptogenic lesion resection, our patient did not accept any invasive procedure.

Conservative management with AED and regular follow-up with an epileptologist are considered in patients unable to be compliant with the surgical treatment, as well as patients with cavernomas that are difficult to get to (i.e., lesion in the brainstem) or contiguous to eloquent brain areas. In concordance, several reports have shown adequate symptomatic response in patients with cavernoma-related epilepsy treated only with AED [30,31]. Conversely, early surgery is considered in patients with a high risk of bleeding with poor adherence to the medication or drug-resistant epilepsy [9]. For instance, Rammo et al. reported a 12-year-old male with refractory epilepsy caused by a right superior temporal cavernoma associated with false lateralization by scalp EEG. In that case, a cavernoma lesionectomy was performed. In addition, FDG-PET images showed bilateral hypometabolism with two irritative zones (frequent spikes) detected by magnetoencephalography (MEG). One of them was surrounded by the cavernoma in the left anterior temporal lobe and the second one in the right intrasylvian region, which was difficult to detected by EEG [10]. Even when findings on MRI and EEG are discordant, surgical intervention may still be offered if ictal onset is definitely present on the intracranial EEG or intraoperative electrocorticography (ECoG), in order to improve prognosis and treatment outcomes in drug-resistant epileptic patients [11,32]. In a study of Jin et al. using MEG and EEG, it was found that patients with a single cavernoma have different spike foci distributions: perilesional, at a remote site in the ipsilateral hemisphere and in a contralateral homologous location (called as “mirror”). Moreover, they reported a higher detectability of mirror and remote spikes by MEG than scalp EEG in patients with single cavernoma and, as in our case, the mirror spike suggests a rapid propagation of the interictal activity from the perilesional epileptogenic focus [33]. In the case of residual epileptic foci after extended lesionectomy and intraoperative ECoG monitoring, additional thermocoagulation could be required. Patients with lesions in surgically deep areas or located in regions of high morbimortality would benefit from stereotactic radiosurgery [15], together with new technologies in surgical neuronavigation, tractography, and awake mapping [34,35,36]. New reports derived from in vitro and in vivo studies suggest that in-depth knowledge in the molecular and cellular mechanism underlying the cavernoma formation may open the possibility for the development of new target inhibitors such as antineoplastic and antiangiogenic drugs as an alternative treatment to surgical and AED management in the case of intractable epilepsy induced by cavernoma malformations [37].

## 4. Conclusions

Most reports of the false lateralization of seizures have been described in patients with temporal epilepsy. Indeed, the few cases of false lateralization of cavernomas reported in the literature correspond to the temporal lobe, including the hippocampus. This is the first case of apparent false lateralization in the temporal region reported in a patient with a cavernoma located in the frontal lobe. Although the exact mechanism of false lateralization is not known, this case suggests a predilection for rapid electrical spread through the frontal system and the hippocampal commissure to the contralateral hemisphere that is difficult to detect by scalp EEG. Considering that cortical lesions in brain MRI do not necessarily indicate the exact region of ictal onset, individuals with discordance between ictal semiology and scalp EEG would benefit from intracerebral recordings and functional neuroimaging, such as MEG or PET. Adequate neurophysiological monitoring and a histopathological study, as well as greater knowledge of the molecular and genetic background, could be useful for developing new therapeutic measures in patients with cavernoma-related epilepsy that does not respond to AED or who cannot have surgery. Although our patient declined surgical workup, including invasive monitoring (a procedure that is considered the gold standard to support the false lateralization diagnosis), we consider that ictal symptomatology provides important information in cases of apparent false lateralization.

## Figures and Tables

**Figure 1 brainsci-10-00584-f001:**
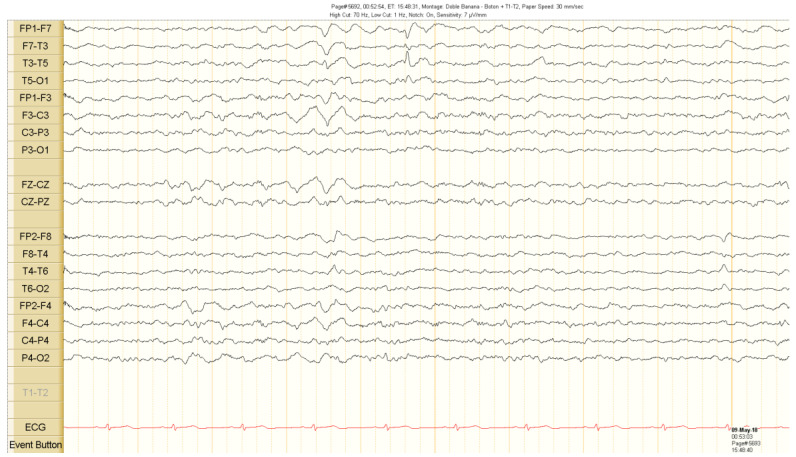
Interictal scalp electroencephalogram (EEG) recordings. Spike activity in the left temporal region during sleep. See the text for detailed description.

**Figure 2 brainsci-10-00584-f002:**
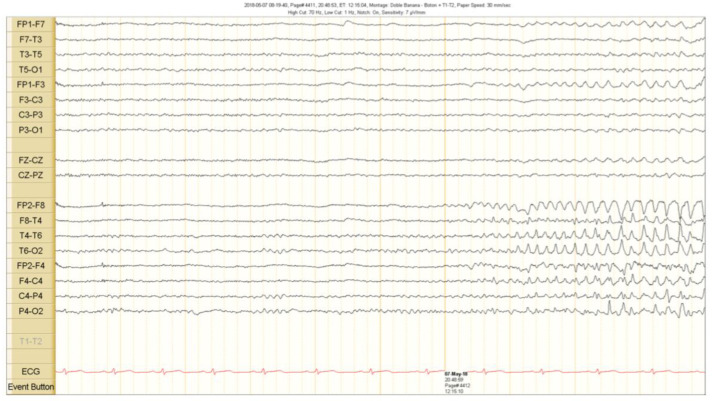
Ictal scalp EEG recordings. A theta activity starts in the right fronto temporal region 5 s before the clinical seizure onset. See the text for detailed description.

**Figure 3 brainsci-10-00584-f003:**
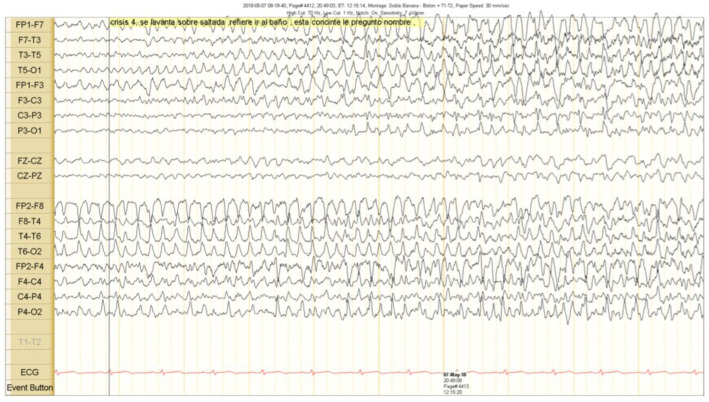
Clinical seizure onset is marked with a vertical black line. The right epileptiform activity spread to the contralateral hemisphere.

**Figure 4 brainsci-10-00584-f004:**
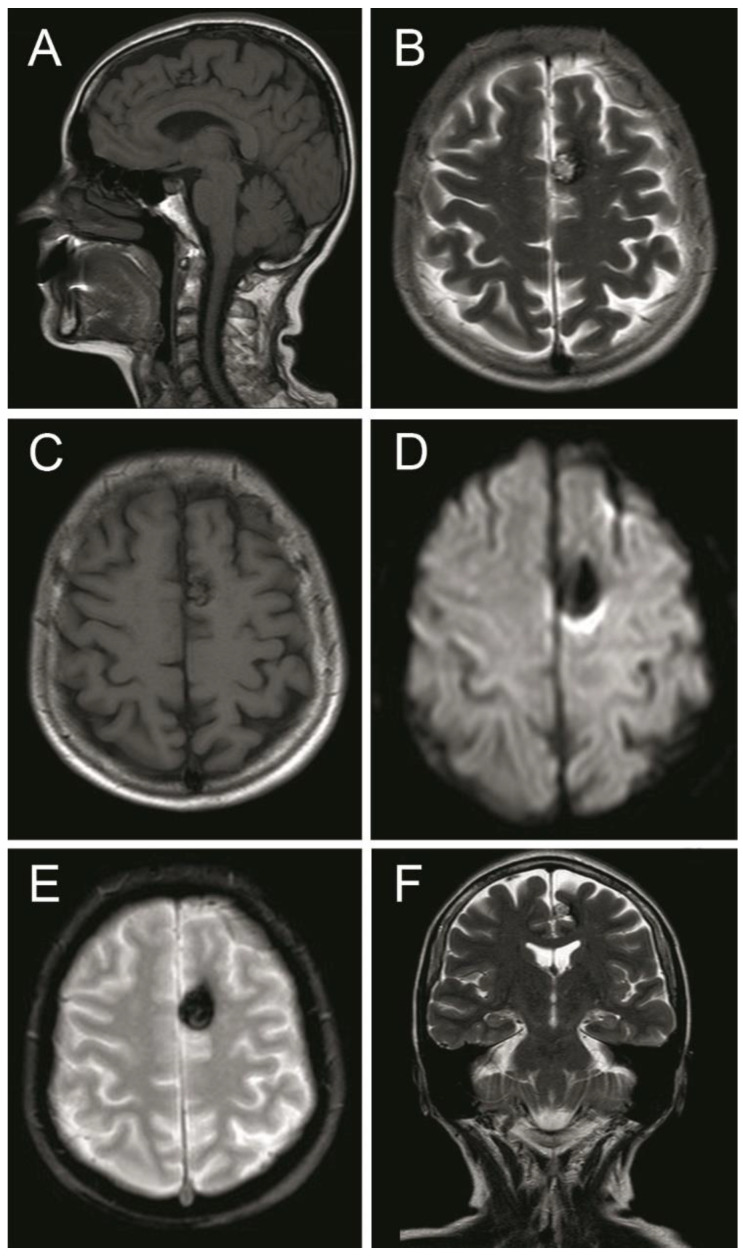
Magnetic resonance image (MRI) sequences showed a left parasagittal cavernoma with a hemosiderin component. (**A**) MRI T1 weighted—sagittal projection. (**B**) MRI T2 weighted—axial projection. (**C**) MRI T1 weighted post Gadolinium—axial projection. (**D**) MRI Diffusion-weighted—axial projection. (**E**) MRI Gradient echo—axial projection. (**F**) MRI T2 Flair weighted—coronal projection. See the text for detailed description.

**Figure 5 brainsci-10-00584-f005:**
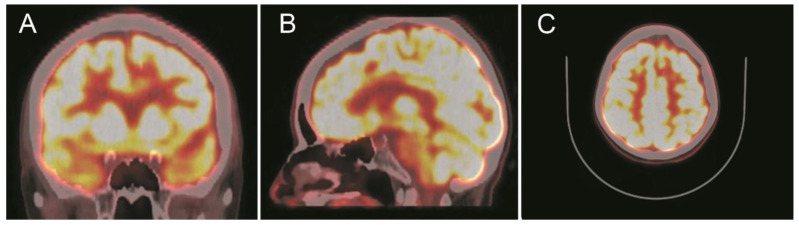
Positron emission tomography with [18] fluorodeoxyglucose (18F-FDG-PET) scans show a focal ametabolic area corresponding in location and size to the frontal cavernoma. (**A**) Coronal projection. (**B**) Sagittal projection. (**C**) Axial projection. See the text for a detailed description.

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
