# Peer review of "Apparent False Lateralization of Seizure Onset by Scalp EEG in Temporal Lobe Epilepsy Associated with Cerebral Cavernous Malformation: A Case Report and Overview"

_brainsci, 2020, doi:10.3390/brainsci10090584_

Round 1

Reviewer 1 Report

This is a very interesting case.

False lateralization of EEG might occur due to

i) intermemispheric activity projecting to the contralateral parasagittal areas or to ii) mirroring or remote propagating activity. In this case, situation

i) is less likely because of temporal involvement. The authors may be interested in an MEG study published by Jin et al (Neuromagnetic localization of spike sources in perilesional, contralateral mirror, and ipsilateral remote areas in patients with cavernoma. Epilepsia 48: 2160-2166, 2007) to consider situation ii).

Author Response

We thank the reviewers for their thorough and constructive comments. Guided by the comments of the reviewers we have now made changes to our manuscript.

We agree with the reviewer that our case would be explained by mirror propagating activity and we include the paper suggested in the discussion section as follows:

“In a study of Jin et al. using MEG and EEG, it was found that patients with a single cavernoma have different spike foci distributions: perilesional, at a remote site in the ipsilateral hemisphere and in a contralateral homologous location (called as “mirror”). Besides, they reported higher detectability of mirror and remote spikes by MEG than scalp EEG in patients with single cavernoma; and as in our case, the mirror spike suggests a rapid propagation of the interictal activity from the perilesional epileptogenic focus”.

We included new figures (figure 2A-2B), in order to describe the temporal relationship between electrical and clinical seizure. 

Reviewer 2 Report

As the authors describe in their manuscript, false localization and false lateralization are common findings in the scalp EEG evaluation of patients with medically intractable epilepsy. However, confirmation of an error in localization or lateralization requires either intracranial EEG monitoring to confirm site of seizure focus onset or lesion resection and subsequent seizure freedom to “prove” the seizure focus localization hypothesis. Although, the authors acknowledge these truths in the manuscript the assertion of false lateralization is an hypothesis until proven.

The authors provide a mini-review of the literature concerning false lateralization and treatment of cavernomas. However, to stand alone as a review paper on one of these topics, the manuscript should be more complete in covering one of these topics.

Figure 5 B. and C. projections should be corrected

Author Response

We thank the reviewers for their thorough and constructive comments. Guided by the comments of the reviewers we have now made changes to our manuscript and we will respond point by point the suggestions and commentaries of the reviewer.

We agree that our case of false lateralization is a hypothesis that has to be proved by intracranial EEG monitoring. Considering that our patient did not accept this procedure, we denominated this case of false lateralization as “apparent”.

This manuscript was sent to Brain Sciences Journal as Case Report and not as  review. Although we performed a mini-review in the discussion section, we changed the title and abstract to avoid this confusion. We included new information regarding the neuromagnetic localization of spikes in patients with cavernomas to improve this section as follows:

“In a study of Jin et al. using MEG and EEG, it was found that patients with a single cavernoma have different spike foci distributions: perilesional, at a remote site in the ipsilateral hemisphere and in a contralateral homologous location (called as “mirror”). Besides, they reported higher detectability of mirror and remote spikes by MEG than scalp EEG in patients with single cavernoma; and as in our case, the mirror spike suggests a rapid propagation of the interictal activity from the perilesional epileptogenic focus”.

We corrected the figure legends 5B and C. We apologize for this mistake. We changed the figures of the EEG recordings (new figures 2A-2B), in order to describe the temporal relationship between electrical and clinical seizure. 

Reviewer 3 Report

The authors describe a patient with seizures due to a left mesial frontal cavernoma. The semiology of the seizures is consistent with the location but the ictal EEG lateralises to the opposite hemisphere. No other lesions have been identified. False lateralisation of EEG and rapid spread EEG in frontal lesions is well described and has been documented with sEEG, although it may not have been described previously with cavernomas.  The finding is unusual but  such discordance is not rare in presurgical evaluation.

The authors do not state the timing of the EEG findings in relation to clinical seizure onset. EEG onset significantly later than clinical onset is recognised to be unreliable. If the onset were early in relation to clinical onset, this would add importance to the finding. If many seconds later, reduce its importance.

The authors do not describe seizures using the 2017 ILAE classification. 

The discussion is rather long and goes into too much detail of areas of peripheral relevance to the core message of the case report.

Author Response

We thank the reviewers for their thorough and constructive comments. Guided by the comments of the reviewers we have now made changes to our manuscript and we will respond point by point the suggestions and commentaries of the reviewer.

Regarding the timing of the EEG findings in relation to clinical seizure onset, we now clarify that epileptiform EEG activity started several seconds early to clinical seizure. We changed the figures 2 and 3 (new figures 2A-2B), in order to describe the temporal relationship between electrical and clinical seizure. 

We changed the description of the seizures according to ILAE 2017 classification as follows:

“The patient was a 53-year-old right-handed female who presented an unknown onset seizure with motor activity (tonic- clonic) at the age of 17 years. Personal history….”.

“This last seizure had a focal onset, with impaired awareness, and a motor component that involved manual automatism, blinking, cephalic version to the right and asymmetric tonic limb posturing with progression to bilateral tonic–clonic seizures.”

“the first type was a seizure with a focal onset, impaired awareness, and non-motor components such as behavior arrest, cognitive impairment due to disorientation and emotional seizures (psychomotor agitation, fear and anxiety) and eye‐blinking; the second one was a focal onset seizure with impaired awareness, motor components such as eye blinking, cephalic version to the right, dystonic extension posture in the right arm, figure of 4, and finally a progression to bilateral tonic- clonic seizure.”.

The discussion was structured in order to provide some context about false lateralization and  cavernoma-related epilepsy following this order:  main clinical findings, hypothesis about the mechanisms of false lateralization; dual pathology hypothesis; paraclinical studies (invasive and non-invasive methods) to study false lateralization; new therapeutical approach to cavernoma-related epilepsy.

According to the suggestions of one the reviewers, we include new information regarding the neuromagnetic localization of spikes in patients with single cavernoma as follows:

“In a study of Jin et al. using MEG and EEG, it was found that patients with a single cavernoma have different spike foci distributions: perilesional, at a remote site in the ipsilateral hemisphere and in a contralateral homologous location (called as “mirror”). Besides, they reported higher detectability of mirror and remote spikes by MEG than scalp EEG in patients with single cavernoma; and as in our case, the mirror spike suggests a rapid propagation of the interictal activity from the perilesional epileptogenic focus”.

Round 2

Reviewer 2 Report

I understand the authors submitted the manuscript as a case report. However, a case of lesional epilepsy with incongruent scalp EEG localization or lateralization is not a novel or useful report. In its current form, the reader does not gain new knowledge about the disease process or treatment approach. If the authors were to develop the manuscript into a thoughtful review of the topic of lesion / EEG incongruence, this could be of value to a reader.  

Author Response

In order to give to the readers an overview of the main findings of false lateralization cases reported in the literature, we now included a table that summarizes their clinical and paraclinical features. In addition, this table included some author´s hypotheses about the different mechanisms of false lateralization proposed. Although false lateralization is not an infrequent pitfall during the scalp EEG interpretation, most of these phenomena have been described in patients with hippocampal sclerosis, cortical atrophy in the temporal lobe or in patients with gross focal cerebral lesions (see table 1). Indeed, there are only two cases reports in patients with false lateralization secondary to cavernoma precisely in the temporal lobe. To our knowledge, the present report is the first case of false lateralization in a patient with a focal lesion in the frontal lobe.

In the present article, we want the reader to keep in mind the pathological mechanisms of seizure spreading in patients with discordance on neuroimaging, scalp EEG and seizure semiology, and also, we would like to highlight the usefulness of clinical semiology.

Reviewer 3 Report

The authors have addressed the concerns of the reviewers and the case now more clearly demonstrates the discordance. The MEG paper cited by another reviewer adds context which increases the interest of the case. I continue to feel that the discussion strays into areas peripheral to the core message of localisation - specifically more general treatment issues cited in lines 188-194 and 207-215

Author Response

In order to give to the readers an overview of the main findings of false lateralization cases reported in the literature, we now included a table that summarizes their clinical and paraclinical features. In addition, this table included some author´s hypotheses about the different mechanisms of false lateralization proposed.

We edited the lines suggested by the reviewer in order to be more direct to the core message, and more related to our case.